# Ion concentration measurement using synthetic microfluidic papers

**Haruka Kamiya**[1], **Hiroki Yasuga**[1,2¤], **Norihisa Miki**[1] *

**1** School of Integrated Design Engineering, Keio University, Yokohama, Kanagawa, Japan, **2** Waseda Research Institute for Science and Engineering, Waseda University, Ookubo, Shinjuku-ku, Tokyo, Japan

¤ Current address: Faculty of Core Research, Ochanomizu University, Otsuka, Tokyo, Japan
* miki@mech.keio.ac.jp

**Data Availability Statement:** All relevant data are within the manuscript and its Supporting Information files.

**Funding:** This work was partly supported by Keio University Ishii-Ishibashi Fund for Education and Research Development and AMED under Grant

## Abstract

Non-invasive diagnosis on biological liquid samples, such as urine, sweat, saliva, and tears, may allow patients to evaluate their health by themselves. To obtain accurate diagnostic results, target liquid must be precisely sampled. Conventionally, urine sampling using filter paper can be given as an example sampling, but differences in the paper structure can cause variations in sampling volume. This paper describes precise liquid sampling using synthetic microfluidic papers, which are composed of obliquely combined micropillars. Sampling volume accuracy was investigated using different designs and collection methods to determine the optimal design and sample collecting method. The optimized protocol was followed to accurately measure potassium concentration using synthetic microfluidic paper and a commercially available densitometer, which verified the usefulness of the synthetic microfluidic papers for precision sampling.

## Introduction

Medical diagnosis that is performed beside a patient and produces results in a rapid manner has been attracting considerable attention, also referred to as Point-of-Care Testing (POCT) [1]. As one of POCT fashions, there is a non-invasive testing performed by a patient oneself through small medical devices. Such devices typically diagnose liquid samples obtained from human bodies such as blood, sweat, saliva, tears and urine. Among them, blood provides more information than the other sample types. Despite the difficulty for patients themselves to access blood without the help of medical professionals, glucose monitoring for diabetic patients is one of the few examples currently used [2, 3]. Sweat also provides some physiological information. Microfluidic devices using colorimetric analysis [4] and biosensors using flexible circuit boards [5] have been developed for measuring metabolites, such as glucose and lactate, and electrolytes, such as sodium and potassium ion, in sweat. Saliva is easy to handle and sample. Several biomarkers associated with mental stress and viral hepatitis are found in saliva [6, 7]. The diagnosis on the saliva depends on the oral cavity that is easily affected by food and drink. Tears have been reported to have a glucose concentration similar to blood, which is refreshed every time a person blinks [8, 9]. Collecting tears is challenging, but sensors for contact lens that will enable continuous monitoring are being developed [10]. Urine tests

Number JP20lm0203130h0001 to Norihisa Miki; and by Grant-in-Aid for Research Activity Start-up and for JSPS Fellows (JSPS KAKENHI Grant number JP19K23498 and 20J00716) to Hiroki Yasuga.

have been used to detect pregnancy [11], diabetes [12], and cancers [13] because urine sampling is readily accessible.

Process of the POCT performed by a patient oneself involves liquid sampling, followed by diagnosis through sensing/measuring. When patients perform the liquid sampling, a method must be simple and, more importantly, able to sample accurately even when patients are unskilled. Given as an example of liquid sampling, sampling urine has been conducted by using a paper strip, which is typically made from filter paper [14, 15]. The filter paper-based strips are low cost, easy to handle and capable of retaining liquid samples by capillary force [16]. However, the cotton fibers that compose the filter papers are randomly intertwined, which may lead to volume variation. Microfabrication can offer structures composed of regular micro-patterns, such as micropillar arrays [17], microchannels [18], and synthetic microfluidic paper (SMP) [19], which are potential in the use for precise sampling. In particular, the SMP which is three-dimensional, regularly ordered microstructures of a photosensitive polymer pumps and maintains liquid volumes with excellent accuracy and repeatability.

This paper describes the usefulness of SMP for precise liquid sampling. Initially, we evaluated sampling precision and retention of sampled liquid using the SMP in comparison to conventional filter papers, some part of which were differently analyzed based on our previous results in [20]. Then, we conducted potassium concentration measurement from aqueous solution as shown Fig 1. In these experiments, we supposed urinalysis for a patient of implantable artificial kidney [21]. The implantable artificial kidney requires monitoring of blood potassium concentration of patient because an abnormal blood potassium concentration can cause heart arrhythmias, cardiac arrest, or mental confusion. Blood potassium concentration can be monitored through the urine since the filtrate, has a similar potassium concentration as that in the blood [22].

## Materials and methods

### Chemicals

The following chemicals were used without further purification: isobutyl acetate, acetone, 2-hydroxyethyl methacrylate (HEMA), isopropyl alcohol (IPA), benzophenone, potassium chloride (KCl) (Wako Pure Chemical Industries, Ltd., Japan) and Irgacure819 (BASF Corp., Germany). Ultra-purified water was prepared using a Millipore system (Direct-Q UV3)

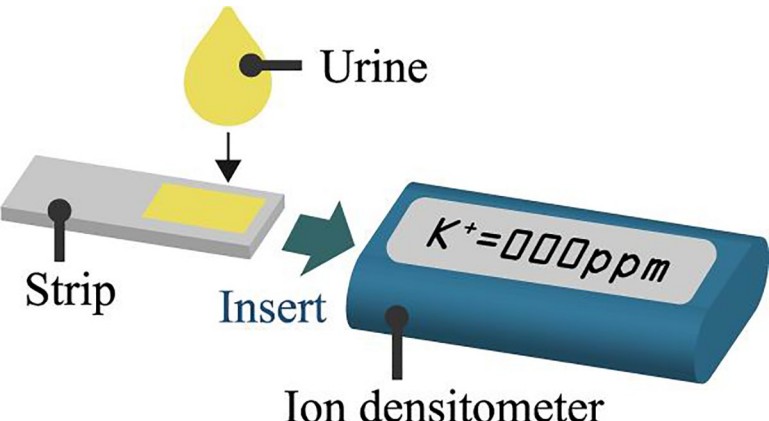

**Fig 1. Conceptual sketch of the ion concentration measurement system.** A precise amount of the urine is sampled by a strip made of synthetic microfluidic paper and the potassium ion concentration is measured using a commercially available ion densitometer.

(Merck KGaA, Germany). A photomask for multidirectional exposure was purchased from Tokyo Process Service Co., Ltd., Japan. Transparent sheets were purchased from Sakae Technical Paper Co., Ltd., Japan. Slide glass (S9111), micro slide glass (S3131), and micro cover glass thickness No. 1 (24×50 mm) were obtained from Matsunami Glass Ind., Ltd., Japan. A potassium ion densitometer (LAQUA twin K-11) was purchased form Horiba, Ltd., Japan. Filter paper used for sampling was Whatman No. 4 (GE Healthcare Japan, Japan).

The sampling strip was based on SMP [19]. The SMP was made of off-stoichiometry thiol-ene (OSTE, OSTEMER 220 Litho, Mercene Labs AB, Sweden). The OSTE is a negative photoresist reported previously [23, 24], and is a viscous liquid at room temperature (15˚C~25˚C). Precursors of OSTE are two monomers, which contains thiol groups and ene (allyl) groups, with a photo-initiator. The thiol-ene reaction triggered by UV exposure induces covalent bonds between the two monomers, resulting in a transparent OSTE polymer [24]. The OSTE contains unreacted thiol groups on the polymer surfaces, which allows surface modification based on polymer-chain grafting [19].

## Fabrication process

SMP made of photosensitive polymer, OSTE, is a paper-like substrate composed of micropillar structures that are combined obliquely, as shown in Fig 2(A). The internal structure of the SMP is characterized by the pillar-to-pillar pitch ($p$), pillar diameter ($d$), pillar height ($h$), and pillar angle ($\theta$). The SMP has many pillars combined with each other and is more robust against collapse than the structures containing vertical straight pillars [19]. This feature enables a pillar matrix with a high aspect ratio ($= h/d$) and a large surface area. These synthetic microfluidic papers allow various flow rates in capillary pumping depending on the structure and have shown smaller deviation than the nitrocellulose-based papers [6].

The SMP was fabricated by multi-directional photolithography, as shown in Fig 3. First, a rectangular-shaped frame with 1-mm thick OSTE was prepared (Fig 3A). The OSTE precursor was poured into this frame, and a photomask was set. As shown in Fig 3B, the precursor was exposed to UV light from four directions through the mask using a collimated UV lamp (EMA-400, Union Optical Co., Ltd., Japan) to form obliquely combined micropillars. A stage containing four slanted aluminum mirrors was used for the multidirectional exposure, as shown in Fig 3B. The OSTE was developed in isobutyl acetate or acetone under ultrasonication (FU-10C, TGK, Japan) (Fig 3C). Then, the surface of the micropillars was hydrophilized by photo-grafting of HEMA (Fig 3D). The entire structure was exposed to UV light for 300 s

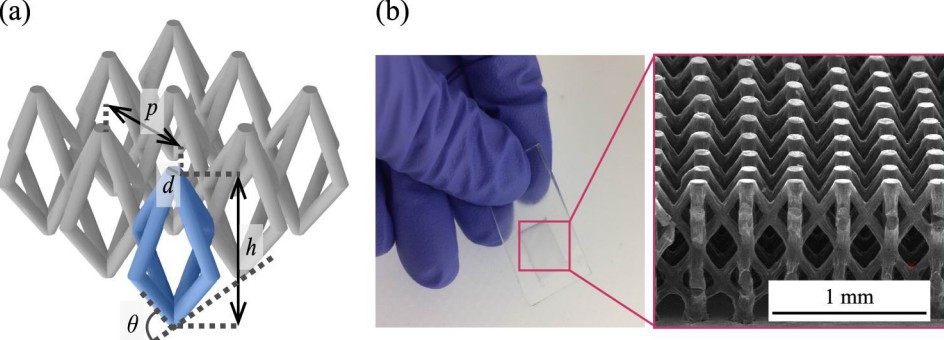

**Fig 2. Internal structure of synthetic microfluidic paper (SMP).** (a) SMP is defined by the pillar-to-pillar pitch ($p$), pillar diameter ($d$), pillar height ($h$) and pillar angle ($\theta$). The value of pitch ($p$) over diameter ($d$) represents pitch ratio ($r$). (b) A photograph of the strip having SMP and a close-up SEM photograph of the SMP.

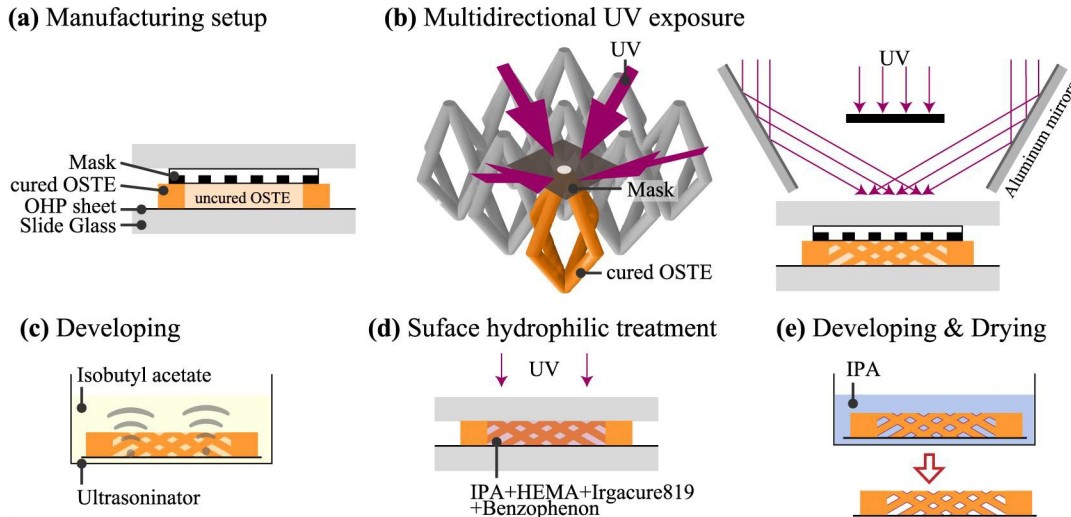

**Fig 3. Fabrication process of synthetic microfluidic paper.** (a) Manufacturing setup; (b) Multidirectional UV exposure to form obliquely combined micro pillars; (c) Development under ultrasonication; (d) Surface hydrophilization; (e) Development and drying.

while immersed into IPA containing HEMA (5 wt%) and the two photopolymerization initiators, Irgracure819 (0.5 wt%) and benzophenone (0.5 wt%). Then, the device was developed again using pure IPA for 300 s, followed by drying (Fig 3E). For this study, six types of SMP with a thickness of 1 mm were prepared (Table 1) with various pitches and diameters. For the experiments, a strip containing the SMP was used, as shown in Fig 2(B).

## Experiments

**Precision sampling experiment.** To optimize the design of the SMP and the method for precise sampling, sampling experiments were conducted with six types of strips (Table 1) using 3 methods: (a) sprinkling method, (b) immersion method I, and (c) immersion method II (Fig 4). 6 types of SMPs were prepared in different fabrication condition as in Table 1, and the SMP in each condition was tested using a same paper strip five times ($n = 5$); sample volume was measured from the weight of the strip before and after sampling. Water was used as the liquid during the experiments. Strip dimensions were 20×10 mm. Detailed procedures for each method are described below.

(a) Sprinkling method: Water was sprinkled onto the strip for 3 s at a rate of 60 mL/min with a smooth flow pump (Tacmina Corp., Japan). This method simulates a patient directly sprinkling urine on a strip to collect a urine sample.

**Table 1. Dimensions of fabricated synthetic microfluidic papers.**

| No. | Diameter ($d$) [μm] | Pitch ($p$) [μm] ($r$ = Pitch/Diameter) |
|:---:|:---:|:---:|
| 1 | 75 | 112.5 (1.5) |
| 2 | 75 | 150 (2.0) |
| 3 | 75 | 225 (3.0) |
| 4 | 100 | 150 (1.5) |
| 5 | 100 | 200 (2.0) |
| 6 | 100 | 300 (3.0) |

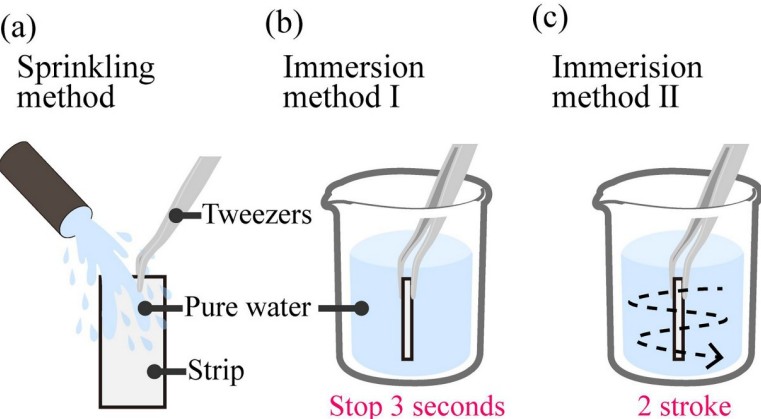

**Fig 4. Precision sampling experiment.** (a) Sprinkling method: Water is sprinkled onto the strip directly. (b) Immersion method I: The strip is immersed for 3 s. (c) Immersion method II: The strip is swirled 2 strokes during a 5-s immersion.

(b) Immersion method I: A strip was placed vertically into a water bath, immersed for 3 s, and then removed. This method simulates a patient immersing a strip into urine collected in a container.

(c) Immersion method II: A strip was placed vertically into a water bath, moved using 2 strokes in 5 s, and then removed. This simulates a patient immersing a strip into urine and swirling the strip before removing it.

All experiments including following experiments were conducted at approximately 23°C and humidity was kept at 50±5%.

**Sample retention test.** As a patient handles a strip, drops of urine could fall off the strip, leading to measurement errors. Therefore, the ability of the strips to retain the urine sample was investigated. During the experiments, a mechanical impact was applied to the strip, as shown in Fig 5. A strip containing a sample was bent by 10 mm, then, it was released abruptly. The weight before and after release was measured. 6 types of SMPs were prepared in different fabrication condition as in Table 1, and the SMP in each condition was tested using a same paper strip five times ($n = 5$).

## Comparison between synthetic microfluidic paper and filter paper

To verify the usefulness of the SMP for precision sampling, we compared SMP with $d = 75$ and $p = 150$ μm and filter paper in terms of the precision sampling tests by immersion method I and sample retention test.

Each strip was cut out into 20×8 mm without any supports and 10 replicates ($N = 10$) were prepared for these tests. In order to avoid the variation of sampling volume, we used vertically

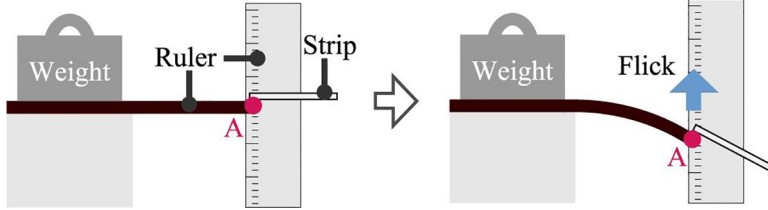

**Fig 5. Sample retention test.** A ruler was fixed and aligned with the edge of the strip. Then, the tip of the strip was bent by 10 mm and released. Weight of the strip before and after the release was measured.

arranged linear actuator that was controlled by microcomputer to pulling strip out of water. Each strip was fixed to a movable stage by stabbing it at the top 1mm by a fishhook-like stainless wire. The pulling speed was set as 10 mm/s. Sample volume was measured from the weight of the strip before and after sampling. After the sampling, sample retention test was conducted sequentially.

**Measurement of potassium ion concentration.** To see if the SMP can be used for measurement system using a paper strip, we conducted potassium concentration measurement using SMP, supposing urinalysis for a patient of implantable artificial kidney. The implantable artificial kidney being developed by our group has a mechanism whereby the filtrate generated by hemofiltration is conveyed to the bladder, turning into urine [21]. Our previous studies showed that this filtrate is equivalent to blood plasma with respect to waste and electrolytes [21]. Since the blood plasma is mostly composed of water and includes the electrolyte component about 1% and its pH does not deviate significantly from 7, we determined using an aqueous potassium chloride solution as a model urine for potassium measurement in this study.

Potassium ion concentration contained in the fabricated strips was measured using a commercially available potassium ion densitometer. The instrument contains an ion selective electrode, which measures specific ion concentration in a sample solution through a potential difference between the ion selective electrode and the reference electrode, as shown in Fig 6(A) [25, 26]. The potassium ion densitometer could accept a sample by direct injection into the instrument sample port or by placing a sample paper immersed with the sample solution into the instrument. Four sampling methods were investigated, as shown in Fig 6(B) ($N$ = 30 for each method): (i) directly injecting the sample solution using a pipette (control experiment); (ii) placing a SMP strip with sample solution into the instrument; (iii) placing a filter-paper strip into the instrument; and (iv) placing a strip (BY046, Horiba, Ltd., Japan) tailored for the densitometer into the instrument. The SMP had $d$ and $p$ values of 75 and 150 μm, respectively. The sample solutions contained potassium ion concentrations of 60, 140, 200, and 360 ppm, which were prepared by dissolving 60, 140, 200, and 360 mg of potassium chloride in 1 L of ultra-purified water, respectively. Less than 10 seconds were taken for each measurement, which resulted from automatic measurement process of the densitometer used.

## Results and discussion

### Precision sampling experiment

Fig 7 shows the coefficient of variation (CV) of sampling volume for each strip, which was calculated as the ratio of the standard deviation to the mean value ($n$ = 5). The actual volume

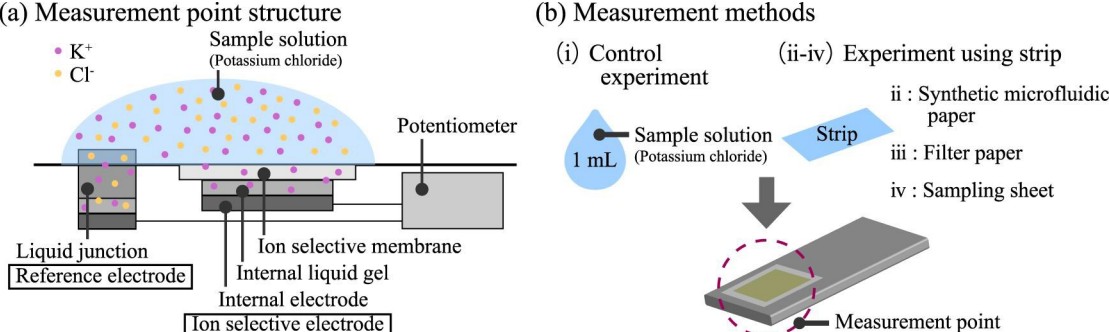

**Fig 6. Measurement of potassium ion concentration.** (a) Measurement mechanism of the densitometer. (b) Measurement methods: (i) Direct injection of the sample solution into the instrument by pipetting; (ii) Placing the synthetic microfluidic paper into the instrument; (iii) Placing the filter paper into the instrument; and (iv) Using the sample strip specific to the densitometer.

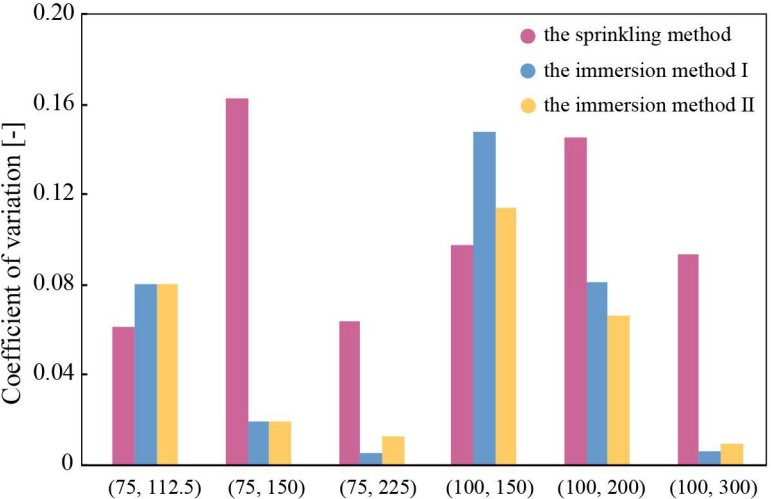

**Fig 7. Results of precision sampling experiments.** The horizontal axis represents sampling strip type and the vertical axis represents coefficient of variation (CV) [–]. The number of trials $n$ using a same paper strip is 5.

sampled by each paper is shown in S1 Fig. The sprinkling method produced no less than 0.06 in CV for all types of SMP. The CV value for immersion methods I and II using SMP decreased as pitch ratio increased, while no significant difference was found between the two methods. Among the conditions tested, immersion method I or II using the SMP with $(d, p)$ of (75, 150), (75, 225), and (100, 300) seemed better for sampling liquid when one needs small CV ($< 0.05$).

## Sample retention test

Fig 8 shows the falling ratio ($R_F$), which represents the ratio of the falling amount ($V_F$) to sampling volume ($V$), as described in Eq (1).

$$R_F = 100 \frac{V_F}{V} \tag{1}$$

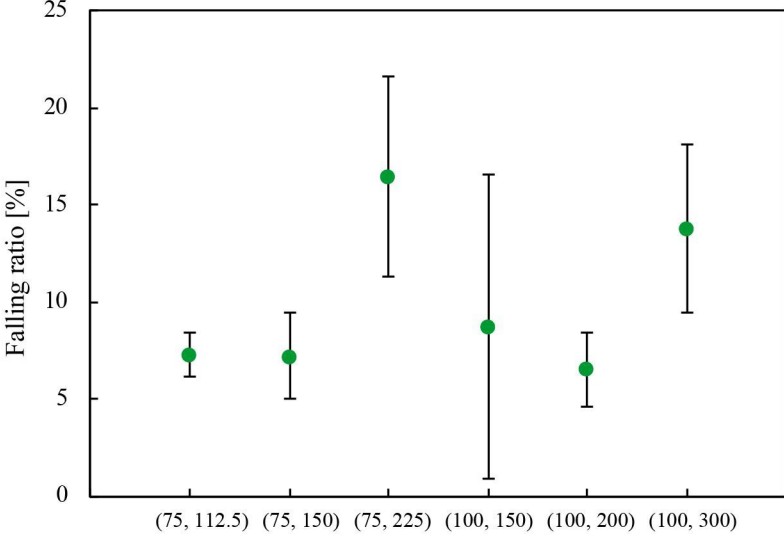

**Fig 8. Results of sample retention tests.** The horizontal axis represents strip type, while the vertical axis represents falling ratio [%]. Error bars represent standard deviation. The number of trials $n$ using same paper strip is 5.

The error bars represent standard deviation. The SMP with a pitch rate of 3.0 had a greater $R_F$ value than other conditions. It is deduced SMP with a large pitch ratio were susceptible to mechanical disturbance. The SMP with $(d, p)$ of (75, 112.5), (75, 150), (100, 150), and (100, 200) had the similar $R_F$ value. It is deduced the SMP were less susceptible to mechanical disturbance, resulting in lower variations in sampling volume. As shown in S1 Table, it is certain that there is a significant difference between these parameters ((d, p) of (75, 112.5), (75, 150), (100, 150), and (100, 200)) and the others, when an α-level of 0.05 was used. On the basis on the results of the precision sampling experiments and sample retention test, we decided the use of the SMP with (d, p) of (75, 150) as a sampling strip for the following experiments.

## Comparison between synthetic microfluidic paper and filter paper

Fig 9(A) shows the coefficient of variation (CV) of sampling volume for the SMP and the filter paper, which was calculated as the ratio of the standard deviation to the mean value ($N = 10$). The actual volume sampled by each paper is shown in S2 Fig. The CV value of the SMP was slightly greater than that of the filter paper. However, we performed F-test on the two normalized data set (ratio of sampled mass over its mean mass) of the SMP and the filter paper by "ftest" function in Excel, and, when an α-level of 0.05 was used, no significant difference was found between them (P = 0.382 > α). Fig 9(B) shows the $R_F$. The error bars represent standard deviation. T-test performed on the two groups by "ttest" function in Excel showed no significant difference between the SMP and the filter paper (P = 0.078 > α), when an α-level of 0.05 was used. On the other hand, F-test performed on the two groups showed that there is a significant difference between variation for the two samples (P = 0.0056 < α), when an α-level of 0.05 was used.

In summary of the comparison between the SMP and the filter paper, they were found to have comparable intra-assay precision in water sampling and retention, but robustness to mechanical disturbances was shown more in the SMP than the filter paper. However, there is a big difference between the SMP and the filter paper: the former was fabricated manually in a

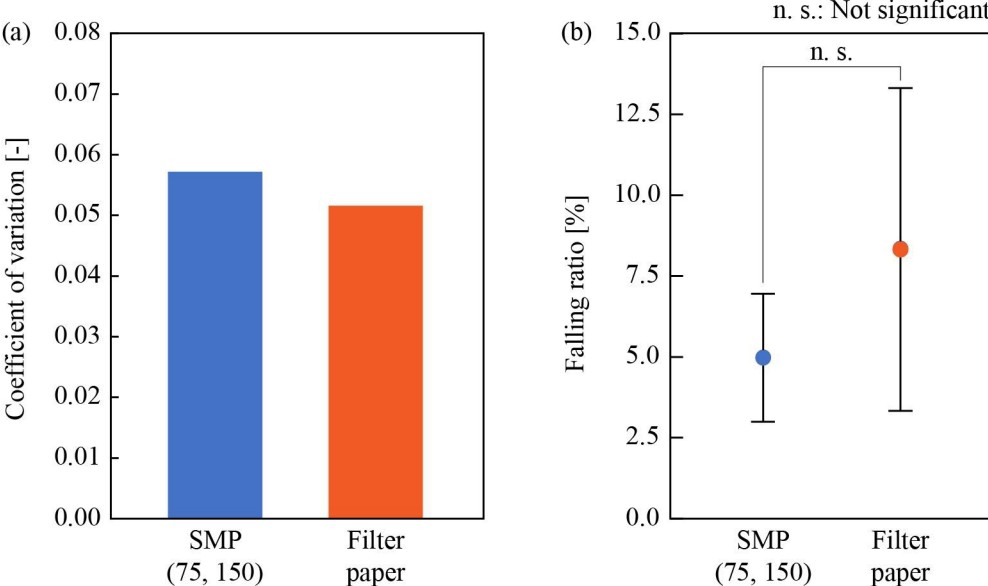

**Fig 9.** Results of precision sampling tests (a), and sample retention tests (b), comparing synthetic microfluidic paper (SMP) and filter paper. The number of trials *N* using different paper strip every time is 10.

laboratory but the latter was done by mass-production process. Actually, the mass variation of the SMP and the filter paper was 0.078 and 0.033 in CV. Automated process or reduction of manual process, if possible, are expected to decrease variation of mass in SMP and consequently variation of sampling volume.

From a future perspective, sampling liquid by SMP has challenges in terms of evaluation for long-term or multi-person use. In this work, the usability of SMP for precise sampling has been confirmed in short-term period. It is also important to know long-term capability to perform precise sampling as a reference for a shelf-life. To get longer-term capability, optimization of polymer formulation for the SMP is one of the issues to be considered. Another challenge to be addressed is multi-operator tests. Depending on the application destination, it is necessary to test with a wide range of subjects who have different age and presence or absence of disease.

## Measurement of potassium ion concentration

Potassium ion concentration in four types of strips was measured using a commercially available densitometer. Fig 10(A) shows the relation between measured potassium ion concentration and the reference values for each measurement method. Fig 10(B) shows the relative error $\eta$, which is described by Eq 2. The value of $C$ represents the measurement result while $C_r$ is the reference concentration of the sample solution. Standard solution potassium ion concentration [ppm] represents the concentration of the sample solution.

$$\eta = 100 \frac{C_r - C}{C_r} \tag{2}$$

For SMP, the relative error $\eta$ was 0.38 to 2.2%, with good accuracy. Because the relative error in the control experiment was −1.2 to 0.48%, measurement using SMP had accuracy

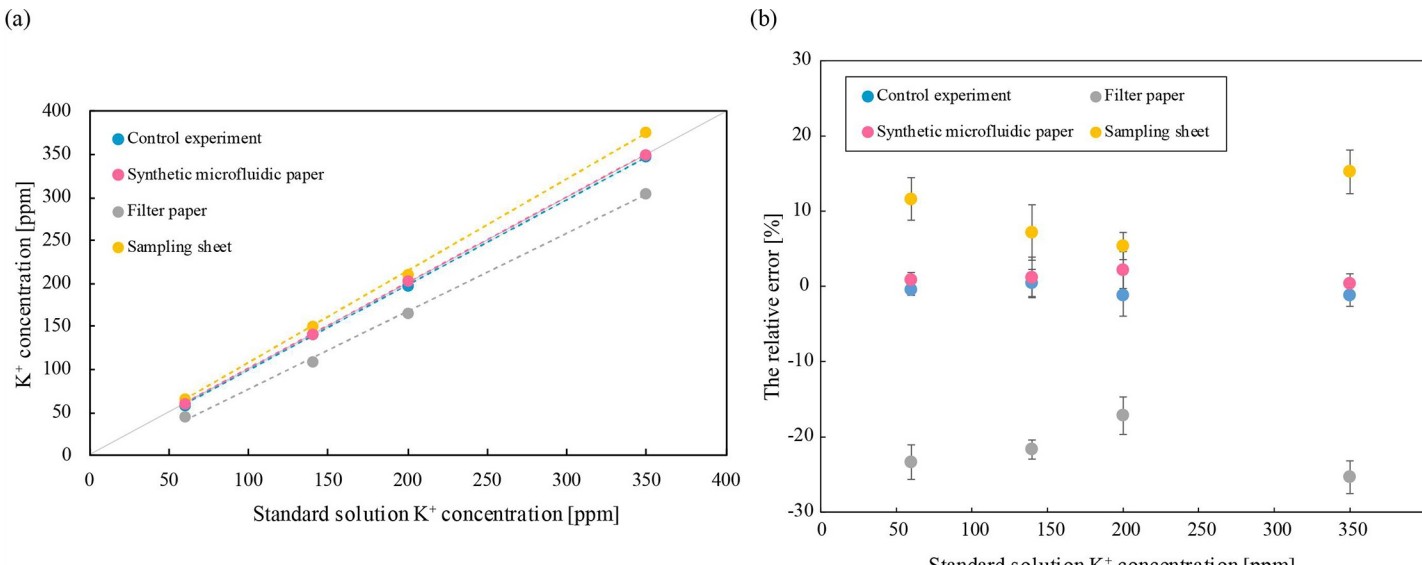

(a)

(b)

**Fig 10.** (a) Potassium ion concentration in different sampling papers and (b) Relative error for potassium ion concentration measurements. (a) The horizontal axis represents standard solution potassium ion concentration [ppm] and the vertical axis represents measured potassium ion concentration [ppm]. (b) The horizontal axis represents standard solution potassium ion concentration [ppm], the vertical axis represents relative error [%], and error bars represent standard deviation. For SMP, the number of trials $n$ using a same strip is 30 and for other paper the number of trials $N$ using different paper every time is 30.

comparable to the control experiment. For measurement using filter paper, the relative error rates were −17 to −25%. For the sampling sheet, we observed the tendency that the measured value was greater than the control experiments. We speculate that the large relative error of concentration measured using filter paper results from the negative charge of the fiber that is the raw material of the filter paper [27]. The results showed that the SMP was better than the filter paper and the sampling sheet provided by the densitometer manufacturer. This result suggests that the SMP is applicable to precision sampling of filtrate of a patient of implantable artificial kidney, followed by the potassium concentration measurement.

## Conclusion

The usefulness of SMPs for the precise collection of liquid samples (such as urine) for ion concentration measurement was investigated. Precise and consistent sampling volumes were achieved with the SMPs being comparable to filter paper when sampling was done by immersion into the liquid. The robustness against mechanical disturbance was high when the SMP with small pitch ratio was used. In the concentration measurement, the strip of SMP was more useful than the filter paper, and the sampling sheet provided by the densitometer manufacturer.

## Supporting information

**S1 Table. P-value derived from T-test on sampling results using synthetic microfluidic paper.**
(TIF)

**S1 Fig. Mass per unit capacity of water sampled by synthetic microfluidic paper (SMP) in precision sampling experiments.**
(TIF)

**S2 Fig. Mass of water sampled by synthetic microfluidic paper (SMP) and filter paper.**
(TIF)

## Author Contributions

**Conceptualization:** Norihisa Miki.

**Data curation:** Haruka Kamiya, Norihisa Miki.

**Formal analysis:** Haruka Kamiya, Hiroki Yasuga.

**Funding acquisition:** Hiroki Yasuga, Norihisa Miki.

**Investigation:** Haruka Kamiya, Hiroki Yasuga.

**Methodology:** Haruka Kamiya, Norihisa Miki.

**Resources:** Hiroki Yasuga, Norihisa Miki.

**Supervision:** Hiroki Yasuga, Norihisa Miki.

**Validation:** Hiroki Yasuga, Norihisa Miki.

**Visualization:** Haruka Kamiya, Hiroki Yasuga.

**Writing – original draft:** Haruka Kamiya, Hiroki Yasuga.

**Writing – review & editing:** Norihisa Miki.

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
