## [Decision Letter · Decision Letter 0]

9 Jan 2020

PONE-D-19-32085

Ion concentration measurement using synthetic microfluidic papers; a new technology of precise sampling

PLOS ONE

Dear Miss Kamiya,

Thank you for submitting your manuscript to PLOS ONE. After careful consideration, we feel that it has merit but does not fully meet PLOS ONE’s publication criteria as it currently stands. Therefore, we invite you to submit a revised version of the manuscript that addresses the points raised during the review process.

We would appreciate receiving your revised manuscript by Feb 23 2020 11:59PM. To enhance the reproducibility of your results, we recommend that if applicable you deposit your laboratory protocols in protocols.io, where a protocol can be assigned its own identifier (DOI) such that it can be cited independently in the future. For instructions see: http://journals.plos.org/plosone/s/submission-guidelines#loc-laboratory-protocols

We look forward to receiving your revised manuscript.

Kind regards,

Albert J. Fornace Jr, MD

Academic Editor

PLOS ONE

Additional Editor Comments:

The authors need to address the concerns of both reviewers.

An administrative issue needs to be addressed about how closely this submission is related to a previously published article in the proceedings of the 40th Annual International Conference of the IEEE Engineering in Medicine and Biology Society (EMBC), https://doi.org/10.1109/EMBC.2018.8513267. Can you confirm that the present work has been written as an independent article, the related works have been adequately discussed by the authors, and that the separation into more than one article has not compromised the robustness of the statistical analysis (e.g., by removing required adjustments for multiple hypothesis testing). For further information on submission guidelines on related manuscripts, please see http://journals.plos.org/plosone/s/submission-guidelines#loc-related-manuscripts.

Reviewers' comments:

Reviewer's Responses to Questions

**Comments to the Author**

1. Is the manuscript technically sound, and do the data support the conclusions?

Reviewer #1: Yes

Reviewer #2: No

2. Has the statistical analysis been performed appropriately and rigorously? 

Reviewer #1: I Don't Know

Reviewer #2: No

3. Have the authors made all data underlying the findings in their manuscript fully available?

Reviewer #1: Yes

Reviewer #2: Yes

4. Is the manuscript presented in an intelligible fashion and written in standard English?

Reviewer #1: Yes

Reviewer #2: Yes

5. Review Comments to the Author

Reviewer #1: The authors present a very clear and simple comparison paper that nicely addresses ion concentration measurement using synthetic microfluidic papers. To better comprehend that information presented the following questions/comments should be addressed:

1) Please describe conditions of testing in more detail, as temperature, humidity, drying time, measurement time can all effect ion concentration measurements using microfluidic paper. The possible condition limitations are important to a medical setting.

2) Please address the appropriateness of using your sample size (n=5).

Reviewer #2: Authors present in this manuscript an interesting microfluidic-based point-of-care device whose paper properties seem to improve sample volume recovery and ion measurement in large samples such as urine. However, additional experiments and precision in methodology would be required to sustain these promising assertions.

Major points:

- Overall, it is difficult to know how many replicates have been used for each experiment. Authors should clarify this point in experiments section and add n numbers in legend of each figures when applicable. In “precision sampling experiment” in experiments section, authors claimed “six samples (N=5, for each) were collected for each method…” but it is unclear. Power analysis is not provided, but I would suggest authors to perform each condition with at least 10-20 replicates.

- Intra- and inter-assay variation could also be an important parameter to assess. I would suggest authors to perform precision sampling and sample retention test experiments on (1) same lot of microfluidic paper and same operator over a short period of time (intra-assay) and (2) different lot over an extended period of time performed by multiple operators (inter-assay).

- For the precision sample experiment, in addition to the coefficient of variation, it would be interesting to also show the actual volume collected by each strip/filter paper (by including errors bars and calculating difference between each method).

- Line 195, please discuss/provide an explanation on why the synthetic microfluidic paper with (d, p) of (75, 150), (75, 225), and (100, 300) was more suitable for producing consistent sampling volume than others.

- Line 206-208, “the values of the synthetic microfluidic papers with a pitch rate of 1.5 and 2.0 and (d, p) of (75, 112.5), (75, 150), (100, 150), and (100, 200) were approximately the same as that of filter paper” and “the synthetic microfluidic paper with a pitch ratio of 3.0 had a greater value”. Is it significant? Authors should calculate significance of difference between strip themselves and filter.

- Time between collection and analysis is also a parameter that can widely change (especially when performed directly by patient) and induce important variation. Therefore, authors should specify the period of time between sample collection and measurements they performed. In addition, I would also suggest authors to evaluate potassium concentration at different time after collection in order to assess if microfluidic strip can result in better sample preservation than filter paper.

- Authors assessed the interest of their device in a context of “urinalysis” for a patient of implantable artificial kidney. Although it is surprising that authors do not use urine sample being given the easy access, they claimed that they used water to test their device. What is the pH of this solution? pH can impact ion measurement and it would have been a minimum to use a solution with a pH equal to urine (~6). Even more interesting, potassium measurement could have been tested with solution with different pH (4-8).

- Line 237, authors claimed “This result suggests that the synthetic microfluidic papers are applicable for precision sampling of urine for the potassium concentration measurement.” This seems overstated since authors did not assess potassium concentration in urine. In addition, sample nature (water spiked with K+?) needs to be specified in Materials and methods section.

- Sensitivity could also have been evaluated by measuring the lowest achievable potassium concentration with the different methods.

Minor points:

- In introduction (line 52), please specify what “unprecise sampling” means (volume variation, stability, concentration variation, etc.?)

- In introduction (line 66), please remove “or urine”

- In “fabrication process” in Materials and methods section, authors claimed that “The synthetic microfluidic paper … is more robust against collapse than the structures containing vertical straight pillars”. Reference (or data) should be included to support this affirmation.

- Please specify how CV has been calculated

6. PLOS authors have the option to publish the peer review history of their article (what does this mean?). If published, this will include your full peer review and any attached files.

Reviewer #1: No

Reviewer #2: No

---

## [Author Response · Author response to Decision Letter 0]

6 Oct 2020

Our response to the two reviewers’ comments on the previously submitted manuscript for PLOS ONE the Response to Reviewers file. In the revised manuscripts, these comments are reflected in the main text and the electronic supporting information.

---

## [Decision Letter · Decision Letter 1]

29 Oct 2020

Ion concentration measurement using synthetic microfluidic papers

PONE-D-19-32085R1

Dear Dr. Kamiya,

We’re pleased to inform you that your manuscript has been judged scientifically suitable for publication and will be formally accepted for publication once it meets all outstanding technical requirements. Both reviewers and I  recommend acceptance.  Minor corrections are needed for misspelling errors: e.g. Fig 4: Immersion method (immerision) and Suppl Fig 1: Y axis title: capacity (capasity).

Kind regards,

Albert J. Fornace Jr, MD

Academic Editor

PLOS ONE

Additional Editor Comments (optional):

Recommend acceptance. Minor corrections are needed for misspelling errors: e.g. Fig 4: Immersion method (immerision) and Suppl Fig 1: Y axis title: capacity (capasity).

Reviewers' comments:

Reviewer's Responses to Questions

**Comments to the Author**

1. If the authors have adequately addressed your comments raised in a previous round of review and you feel that this manuscript is now acceptable for publication, you may indicate that here to bypass the “Comments to the Author” section, enter your conflict of interest statement in the “Confidential to Editor” section, and submit your "Accept" recommendation.

Reviewer #1: All comments have been addressed

Reviewer #2: All comments have been addressed

2. Is the manuscript technically sound, and do the data support the conclusions?

Reviewer #1: Yes

Reviewer #2: Yes

3. Has the statistical analysis been performed appropriately and rigorously? 

Reviewer #1: Yes

Reviewer #2: Yes

4. Have the authors made all data underlying the findings in their manuscript fully available?

Reviewer #1: Yes

Reviewer #2: Yes

5. Is the manuscript presented in an intelligible fashion and written in standard English?

Reviewer #1: Yes

Reviewer #2: Yes

6. Review Comments to the Author

Reviewer #1: Authors have adequately addressed all comments and have provided extra trials giving the experiments the robustness that was advised.

Reviewer #2: All comments have been addressed.

Minor corrections are misspelling errors. Ex:

Fig 4: Immersion method (immerision)

Suppl Fig 1: Y axis title: capacity (capasity)

7. PLOS authors have the option to publish the peer review history of their article (what does this mean?). If published, this will include your full peer review and any attached files.

Reviewer #1: No

Reviewer #2: No

---

## [Editor Report · Acceptance letter]

9 Nov 2020

PONE-D-19-32085R1 

Ion concentration measurement using synthetic microfluidic papers 

Dear Dr. Kamiya:

I'm pleased to inform you that your manuscript has been deemed suitable for publication in PLOS ONE. Congratulations! Your manuscript is now with our production department. 

Kind regards, 

on behalf of

Dr. Albert J. Fornace Jr 

Academic Editor

PLOS ONE